# Towards Consistent Language Models Using Controlled Prompting and Decoding

## Jasmin Mousavi, Arash Termehchy

mousavij@oregonstate.edu, termehca@oregonstate.edu

## Abstract

Large language models (LLMs) have shown unprecedented abilities in generating linguistically coherent and syntactically correct natural language output. However, they often return incorrect and inconsistent answers to input questions. Due to the complexity and uninterpretability of the internally learned representations, it is challenging to modify LLMs such that they provide correct and consistent results. To address this challenge, recent research has focused on controlling the outputs of LLMs through methods like constrained optimization and probabilistic inference. While these approaches mark significant progress, they have limitations in terms of usability, efficiency, and linguistic coherence. Some methods require extensive fine-tuning, making them less practical for general use, while others compromise the linguistic quality of the output. In order to address these limitations, we explore adding constraints to the prompt. Our experimental findings reveal that this approach significantly reduces the need for model fine-tuning and enhances the quality of the outputs, leading to improvements in efficiency and in the linguistic coherence of the generated output. These findings highlight the importance of end-to-end solutions, where prompts and decoders work together in addressing inconsistencies in LLMs.

## 1 Introduction

Large language models (LLMs) have shown unprecedented abilities in processing natural languages (Radford et al. 2018; OpenAI 2023). They effectively generalize to perform various tasks with few or no training examples. Thus, there is a rapidly growing interest in using them to solve data-driven problems, such as, interactive question answering.

Nonetheless, LLMs often provide incorrect answers to input queries and perform inaccurate inferences (Ji et al. 2023; OpenAI 2023). Several studies indicate the recent LLMs provide up to 40% erroneous answers to factual questions (OpenAI 2023). These erroneous results are important obstacle for use of LLMs in real-world applications.

To address the problem of inaccurate answers returned by LLMs, we should recognize that **LLMs are not knowledge bases, but rather probabilistic or approximate models of factual information**. LLMs may over-generalize patterns and relationships observed in the sub-sequences of pretraining documents, which might lead to returning spurious relationships and inaccurate results. The uninterpretable mixture of linguistic patterns and factual information has made it challenging to eliminate incorrect information.

One approach is to augment LLMs with *additional and potentially relevant information* from external data sources (Liu, Yogatama, and Blunsom 2022; Borgeaud et al. 2022; Mialon et al. 2023), i.e., retrieval-based LLMs. These methods often add extra information to the context considered during pretraining. This line of research have improved the accuracy of LLMs to a limited degree, as it does not address the core issue of having spurious and incorrect information in LLMs. It is unclear whether adding more relevant information eliminate inaccurate information stored in the model. Moreover, finding sufficiently many relevant data sources, particularly for long-tail entities, may pose challenges.

It is challenging to ensure that an LLM learns accurate generalizations and returns correct answers as it may require perfect knowledge of unobserved data. Even with perfect knowledge of unobserved data, it is challenging to guarantee that LLMs learn accurate generalizations and returns correct answers. Nevertheless, we may be able to restrict its decoding to **adhere to declarative constraints** in the domain to avoid generating incorrect results.

Constraints are essentially rules or guidelines that govern the behavior or output of a system. They can be defined by human experts or learned from data in an unsupervised manner (Papenbrock et al. 2015; Baskaran et al. 2017). Compared to retrieval-based augmentation, we argue that **constraints offer a more robust and adaptable framework for reducing inconsistencies in LLMs**. There are two key advantages to using constraints. First, their ability to encapsulate rules governing the underlying domain enables a system to generalize beyond particular instances in a dataset, i.e., out of distribution generalization. Second, that constraints are a form of high-level knowledge, effectively abstracting large quantities of data. Their compressed representation offers a flexible and efficient method of augmenting LLMs by (1) allowing for soft incorporation of constraints (e.g. adhere to a constraint with 80% probability), (2) reducing the size of information used as context to LLMs, and (3) providing a structured way to control the output of LLMs.

There has been recent effort on limiting the output of LLMs so they *follow given constraints*, e.g., contain certain keywords (Lu et al. 2021; Lew et al. 2023). These methods use constrained optimization or probabilistic inference over the sequences generated by the LLM to reduce the probability of the outputs with invalid patterns. These efforts are steps in the right direction but fall short of ensuring usable, scalable, and linguistically coherent outputs from LLMs. For instance, NeuroLogic (Lu et al. 2021) requires task-specific fine-tuning, which is impractical as LLMs grow in size. On the other hand, Sequential Monte Caro (Lew et al. 2023) is compatible with off-the-shelf models, but often fails to maintain linguistic coherence due to its simple masking decoding strategy. Both methods, applied only during decoding, don't address the LLM's potential to learn and represent spurious relationships. This is hard to control due to the difficulty in interpreting LLMs' learned representations. For instance, the learned spurious relationship about one entity might impact how an LLM answers a question about a different but related entity.

To overcome these limitations, we augment the prompt with domain-specific constraints. Prompting with constraints offers three advantages. First, it leverages LLMs' in-context learning capabilities, thereby eliminating the need for fine-tuning. Second, it can introduce domain knowledge not present in the training data, e.g., each patient is a human. Hence, the modified prompt might convey more information about the domain than the original one. Third, it expresses the properties of entities that are consistent with constraints in the domain. Moreover, consistent answers depend on the context of the domain constraints, i.e., different domains require different lines of reasoning. By incorporating these constraints into the input context, LLMs can generate higher quality output distributions, enabling the decoder to work more effectively.

In this paper, we explore the use of constraints within prompts to improve the limitations of NeuroLogic (Lu et al. 2021) and Sequential Monte Carlo (Lew et al. 2023) decoding strategies. We conduct empirical results for integrating constraints in Llama-2 (Touvron et al. 2023) on the CommonGen benchmark (Lin et al. 2019), without fine-tuning (Section 6). We identify and discuss the trade-offs between generation quality, constraint satisfaction, and efficiency. We find that optimizing all these aspects is not possible by just adding constraints to the prompt or decoder alone. However, an end-to-end approach, combining constraints in both prompting and decoding, shows improvement. Specifically, compared to using only decoder strategies, adding constraints to prompts leads to improvements in efficiency and generation quality. These results underscore the effectiveness of end-to-end strategies, where prompts and decoders work together in addressing inconsistencies in LLMs.

## 2 Background

**Constraints.** In our problem, a constraint is defined over the sequence of tokens, i.e., words, generated by an LLM. Given a generated sequence of words $S$, let us define an indicator function $I(w_j, S)$ that returns true if a word $w_j$ occurs in $S$.

A set of constraints can be formulated in Conjunctive Normal Form (CNF) as a conjunction ($\wedge$) of clauses $\wedge_i^n C_i$, where each clause $C_i$ is a disjunction ($\vee$) of literals:

$$\underbrace{(I(w_{11}, S) \vee \cdots \vee I(w_{1k_1}, S))}_{C_1} \wedge \cdots \wedge \underbrace{(I(w_{n1}, S) \vee \cdots \vee I(w_{nk_n}, S))}_{C_n}$$

where each constraint $I(w_j, S)$ represents a literal.

For example, suppose we would like to generate a sentence that uses the concepts from the set of keywords $x =$ [dog, run, field]. Therefore, the objective is to generate an output sequence $S$ that contains all keywords in $x$ or its inflections (e.g., dog = [dog, dogs, dogging, dogged]). This expressed in CNF is:

$$(I(\text{dog}, S) \vee I(\text{dogs}, S) \vee I(\text{dogging}, S) \vee I(\text{dogged}, S))$$
$$\wedge(I(\text{run}, S) \vee I(\text{runs}, S) \vee I(\text{running}, S) \vee I(\text{ran}, S))$$
$$\wedge(I(\text{field}, S) \vee I(\text{fields}, S)) \tag{1}$$

**Constraint Satisfaction.** Given a set of constraints $C$ expressed in CNF and sequence $S$, constraint satisfaction refers to the process of checking if sequence $S$ violates constraints in $C$. This process is gauged using two key metrics: coverage and satisfaction. Coverage is a number between 0 and 1, calculated as the proportion of clauses in $C$ that evaluate to true:

$$\frac{1}{|C|} \sum_{i=1}^{|C|} C_i$$

where $|C|$ is the number of clauses in $C$. Satisfaction is a Boolean assignment (0 or 1) that assesses whether $S$ adheres to *all* clauses in $C$:

$$\wedge_{i=1}^{|C|} C_i$$

For example, given sequence $S$ = "The dogs are in the field" and CNF from equation 1, let us define clause $C_1$ for the clause containing "dog", $C_2$ for "run", and $C_3$ for "field". Here, "dogs" meets the constraint for clause $C_1$, "field" for $C_2$, but $C_3$ is not satisfied. Therefore, the coverage is $\frac{2}{3}$, while satisfaction is 0, as not all constraints are met.

## 3 Problem Formulation

**Input.** We are given pre-trained language model $LM$, a set of constraints $C = [c_1, c_2, ...c_n]$, and prompt $P$. In our problem, $P$ is the prompt that describes the downstream task and can be further modified with a set of in-context few-shot examples.

**Output.** The output is a sequence of tokens, i.e. words, $S$, generated by language model $LM$

**Metrics.** In our problem we consider three metrics: *constraint satisfaction*, *generation quality*, and *time*. As described in Section 2, constraint satisfaction $CS(S, C)$ refers to the coverage and total satisfaction with respect to constraints in $C$ and sequence $S$. Generation quality $GQ(S)$, evaluates the linguistic coherence and fluency of $S$. This evaluation can be conducted through human evaluation or automated metrics, e.g., ROUGE (Lin and Hovy 2003) or

BLEU (Papineni et al. 2002). These automated metrics compare $S$ to a human-generated reference sequence, calculating a quality score (usually ranging from 0 to 1). The scoring is based on specific heuristics, such as the use of n-grams or similarity measures. Time $T(S)$, quantifies efficiency, i.e., the computational time required to generate sequence $S$ (e.g. inference time).

**Problem Definition.** Given a pre-trained language model $LM$, a set of constraints $C = [c_1, c_2, ...c_n]$, and prompt $P$, the objective is to generate a sequence $S$ that maximizes constraint satisfaction $CS(S, C)$ and generation quality $GQ(S)$, while minimizing time $T(S)$.

# 4 Current Approaches for Decoding with Constraints

Due to the uninterpretability of the learned representations in a language model, it is difficult to guarantee that augmenting the prompt with constraints will produce consistent models that return accurate answers. To address this challenge, one may augment the decoder, such that constraints are leveraged during text generation.

**Challenges.** Due to the autoregressive nature of language model $LM$, i.e. $Pr_{LM}(s_{t+1}|s_{1:t})$, it is often expensive to check constraint satisfaction as sequence $S$ is generated. Hence, it is challenging to mitigate this trade-off between constraint satisfaction $CS(S, C)$ and inference time $T(S)$. Another challenge stems from the dramatic distribution changes induced by conditioning $LM$ on constraints $C$ during generation, i.e., $Pr_{LM}(s_{1:N}|C)$. This introduces another trade-off between constraint satisfaction $CS(S, C)$ and generation quality $GQ(S)$.

**Objective.** The objective is to generate a sequence $S$ that maximizes constraint satisfaction $CS(S, C)$, by conditioning the autoregressive generation process of language model $LM$ on constraints $C$, i.e. $Pr_{LM}(s_{1:N}|C) = \Pi_t^N Pr_{LM}(s_{t+1}|s_{1:t}, C)$, while minimizing time $T(S)$ and maximizing generation quality $GQ(S)$.

## 4.1 Decoding Strategies

We leverage two decoding strategies with varying levels of satisfaction: soft constraint decoding with NeuroLogic (Lu et al. 2021) and hard constraint decoding with Sequential Monte Carlo (Lew et al. 2023).

**NeuroLogic (NL)** (Lu et al. 2021) is an inference time decoding algorithm that uses a variant of beam-search. The objective is to optimize the probability of generating sequences while also steering towards constraints using a penalty term. Due to the interest of using off-the-shelf models, we chose *not to fine-tune* an LLM for using the NeuroLogic decoder. It is important to note, however, their experiments were conducted using a fine-tuned model.

**Sequential Monte Carlo (SMC)** (Lew et al. 2023) is an inference time masked decoding algorithm. They model sequence generation as a probabilistic inference problem using a variant of Sequential Monte Carlo with particle filtering. In SMC, a user writes a program that specifies the desired constraints in a sequential manner. The user may also specify the number of particles used, where each particle acts as a weighted sample of the posterior distribution. We programmed constraints, i.e. contains certain keywords, as an infilling problem, where keywords are sampled with a masked vocabulary.

# 5 Prompting with Constraints

Integrating structured data, such as constraints, into prompts poses challenges for LLMs, as they have been trained on unstructured data. It is unclear whether the LLM's low-dimensional representation can accurately and reliably reflect these domain-specific constraints. Hence, it is challenging to ensure constraint satisfaction solely through prompt augmentation. Nonetheless, prompting with constraints can still bias the output distribution, allowing the decoder to work more effectively.

**Challenges.** In our problem, we are trying to augment the prompt $P$ to language model $LM$ with constraints $C$. However, we face the challenge of the limited input context length $k$, e.g., 4096 tokens for Llama-2 (Touvron et al. 2023). This limitation necessitates an optimization approach for prompting, focusing on the effective inclusion and representation of constraints within this restricted token space.

**Objective.** The optimization objectives for prompting are twofold: first, to minimize the textual representation of constraints $C$, i.e., level of abstractness, necessary for effectively guiding the $LM$ to comply with $C$ for prompt $P$ while occupying minimal token space. Second, minimizing the number of constraints used in the input, ensuring the total input length, including prompt $P$ and constraints $C$, does not exceed context length $k$.

## 5.1 Prompting Strategies

For input prompt $P$ = "write a sentence", we construct two prompting techniques for constraints with varying representation sizes.

**Conjunctive Normal Form (CNF)** prompting style models constraints, i.e., keywords, in conjunctive normal form. For example, the keywords [dog, run, field] in conjunctive normal form is (dog ∨ dogs ∨ ... ) ∧ (run ∨ running ∨ ... ) ∧ (field ∨ fields). We can translate this constraint to text by converting ∨ to *or* and ∧ to *and*. Hence, our final prompt is "Write a sentence using the words (dog or dogs or ... ) and (run or running or ... ) and (field or fields)".

**Abstract (ABS)** prompting style describes an abstract instance of a constraint, e.g., "Given a set of words $x$, write a sentence using all words in $x$ or inflections of $x$". Since *ABS* prompts do not include specific instances of keyword inflections, it is more compressed than *CNF* prompts.

## 5.2 Prompting and Decoding with Constraints

Augmenting either the prompt or the decoder in isolation can help reduce inconsistencies in LLMs, but each approach has its drawbacks. While decoders can offer satisfaction guarantees, they may slow down inference and reduce generation quality due to significant shifts in output distribution. In contrast, prompting maintains generation quality but lacks the

ability to ensure constraint satisfaction. Decoder only strategies fail to tackle the LLM's tendency to learn and represent spurious relationships. For example, a misleading association learned about one entity could affect responses about a related entity. Prompting can mitigate this by providing extra context, thereby offering a more comprehensive understanding of the domain and aligning entities with domain constraints. Likewise, decoders can enhance prompts by ensuring constraint satisfaction. To overcome the limitations of each method individually, we propose exploring an end-to-end approach, integrating constraints in both prompting and decoding. Additionally, we aim to investigate how various prompting techniques impact the effectiveness of decoder strategies.

# 6 Empirical Results

In this section we present our empirical results for integrating constraints with LLMs using the CommonGen benchmark (Lin et al. 2019). We identify the risks and trade-offs of augmenting LLMs with constraints for the *prompt only* and *decoder only* strategies, in terms of generation quality, constraint satisfaction, and time.

*Prompt only* strategies include Conjunctive Normal Form (*CNF*) and Abstract (*ABS*) style prompting (Section 5.1). We also supply the prompt with additional in-context examples, i.e. *0-shot*, *1-shot*, and *2-shot*. Examples were extracted from the training set. *Decoder only* strategies include NeuroLogic (*NL*) (Lu et al. 2021) and Sequential Monte Carlo (*SMC*) (Lew et al. 2023) (Section 4.1).

We also explore whether injecting constraints into both *prompt + decoder*, will help or hurt any risks and trade-offs that exist in the prompt or decoder alone.

## 6.1 LLM Implementation Details

We use Llama-2 (Touvron et al. 2023) as our pretrained language model across all experiments. Llama-2 was pretrained over 2 trillion tokens of data between January 2023 and July 2023. Llama-2 consists of 7 billion parameters, 32 layers, 4096 hidden representation size, 32 attention heads, a 4096 token context window size.

## 6.2 Dataset

The CommonGen dataset (Lin et al. 2019) is a benchmark designed for controlling language model generation with constraints, i.e., contain certain keywords. Given a set of keywords, e.g., "dog run field", the goal is to generate a sentence using all the keywords or the infections of the keywords, e.g., "dogs" or "dog". Each set contains a minimum of 3 keywords and a maximum of 5 keywords. The dataset is split into train, validation, and test sets of sizes 64.7k, 4.02k, and 1.5k, respectively. Typically, those using the CommonGen dataset would first fine-tune their language model using the training set. However, given the size of modern LLMs, users may not have the resources for fine-tuning an LLM. Hence, we focus on using inference-based algorithms that can be used with off-the-shelf models *without fine-tuning*. Our results are conducted over the test set.

**Constraints.** In CommonGen, constraints can be defined for a set of keywords $[w1, w2, w3]$ as follows. If $S$ is a sentence, then $S$ must contain $w1$ or one of its inflections, $w2$ or one of its inflections, and $w3$ or one of its inflections. The objective is to generate sentences that adhere to this constraint. A key characteristic of this constraint is its allowance for multiple valid outputs, stemming from the underspecificity of the input. This leads to a wide array of possible sentences that represent instances of the constraint.

**Data Leakage.** Recently, researchers have been concerned with data leakage in LLMs (Carlini et al. 2021, 2022). Due to their ability to memorize training data (Carlini et al. 2022), benchmark performance is often inflated. Given that Commongen is a public dataset, it is likely that Llama-2 has seen this dataset and even memorized ground truth sentences for the train and validation sets. However, since ground truth sentences for the test set are not publicly available, it is unlikely that Llama-2 memorized them.

## 6.3 Metrics

**Generation Quality** is measured using automatic metrics, such as ROUGE (Lin and Hovy 2003), BLEU (Papineni et al. 2002), CIDEr (Vedantam, Zitnick, and Parikh), and SPICE (Anderson et al. 2016). These metrics generate a quality score for the generated sentence based on human generated reference sentences, where a perfect score is 100.

ROUGE-L is a precision and recall based metric that identifies the longest common co-occurring n-grams and sentence-level similarity by calculating the weighted harmonic mean. BLEU-4 is a precision based metric that counts the matching 4-grams between the generated and reference sentences. CIDEr is a consensus based metric that takes the average cosine similarity of Term Frequency - Inverse Document Frequency weighted n-grams. SPICE is a semantic propositional based metric that establishes syntactic dependencies between words, then maps the syntactic dependencies using logical rules, and finally computes the F-score defined over the logical rules.

**Constraint Satisfaction** measures the method's ability to fully satisfy the constraint (used all keywords or their inflections), i.e., *satisfied*. We also calculate *coverage*, which is an average over the percentage of keywords (or their inflections) used in the generated sequence.

**Time** is computed as the time taken (in seconds) for generating a sequence, i.e., inference time.

## 6.4 Results & Analysis

Results over all experiments can be found on Table 1. For a single concept set, we provide the sentence generated across all experiments in Appendix A.

**Prompt Only.** We aim to understand how varying the constraint representation in the prompt, i.e., *ABS* vs. *CNF* and in-context examples, i.e., *n-shot*, impact generation quality, constraint satisfaction, and time.

Across most experiments ABS prompting achieves higher satisfaction than *CNF* prompting. This suggests that LLMs can understand abstract, high-level descriptions of constraints. Given the fact that *CNF* prompts include all the

| Method | Generation Quality | | | | Constraint Satisfaction | | Time |
|---|---|---|---|---|---|---|---|
| | ROUGE-L | BLEU-4 | CIDEr | SPICE | Coverage | Satisfied | Seconds |
| *Prompt Only* | | | | | | | |
| ABS 0-shot | 25.37 | 06.29 | 04.34 | 13.81 | 51.61 | 18.84 | 01.56 |
| ABS 1-shot | 29.46 | 08.41 | 06.22 | 18.49 | 74.49 | 35.34 | 01.85 |
| ABS 2-shot | **31.34** | **10.60** | **07.33** | **20.06** | 76.74 | 38.74 | 01.83 |
| CNF 0-shot | 22.82 | 03.74 | 02.37 | 12.13 | 42.17 | 11.09 | 01.84 |
| CNF 1-shot | 29.60 | 08.07 | 05.88 | 18.77 | **77.47** | **38.88** | 01.51 |
| CNF 2-shot | 30.93 | 09.92 | 06.81 | 19.22 | 74.48 | 34.34 | 01.46 |
| *Decoder Only* | | | | | | | |
| NL (Lu et al. 2021), beam=8 | 10.05 | 00.00 | 00.08 | 02.67 | 02.41 | 00.00 | 03.46 |
| NL (Lu et al. 2021), beam=32 | 10.36 | 00.00 | 00.06 | 02.31 | 01.12 | 00.00 | 12.47 |
| NL (Lu et al. 2021), beam=64 | 9.74 | 00.23 | 00.04 | 02.59 | 00.96 | 00.00 | 24.01 |
| SMC (Lew et al. 2023), particle=8 | 23.10 | 02.60 | **01.71** | 15.37 | **100.0** | **100.0** | 22.92 |
| SMC (Lew et al. 2023), particle=16 | 22.86 | 02.52 | 01.62 | **15.55** | **100.0** | **100.0** | 22.96 |
| SMC (Lew et al. 2023), particle=32 | **22.92** | **02.64** | 01.69 | 15.26 | **100.0** | **100.0** | 23.17 |
| *Prompt & Decoder* | | | | | | | |
| ABS 0-shot + NL (Lu et al. 2021), beam=8 | 36.54 | 14.82 | 10.72 | 20.65 | 95.93 | 83.43 | 05.30 |
| ABS 1-shot + NL (Lu et al. 2021), beam=8 | 39.07 | 19.25 | 12.13 | 23.25 | 94.13 | 76.55 | 04.61 |
| ABS 2-shot + NL (Lu et al. 2021), beam=8 | 39.39 | 19.76 | 12.26 | 23.65 | 93.81 | 75.48 | 05.11 |
| CNF 0-shot + NL (Lu et al. 2021), beam=8 | 15.41 | 03.98 | 01.56 | 06.42 | 09.38 | 01.00 | 07.47 |
| CNF 1-shot + NL (Lu et al. 2021), beam=8 | 39.66 | **25.73** | **13.30** | **24.18** | 79.91 | 39.08 | 09.37 |
| CNF 2-shot + NL (Lu et al. 2021), beam=8 | **39.81** | 25.35 | 12.84 | 23.57 | 75.49 | 27.99 | 11.30 |
| ABS 0-shot + SMC (Lew et al. 2023), particle=8 | 25.86 | 04.00 | 02.79 | 18.80 | **100.0** | **100.0** | 25.33 |
| ABS 1-shot + SMC (Lew et al. 2023), particle=8 | 27.86 | 05.66 | 04.46 | 20.27 | **100.0** | **100.0** | 25.14 |
| ABS 2-shot + SMC (Lew et al. 2023), particle=8 | 28.62 | 06.17 | 04.90 | 20.55 | **100.0** | **100.0** | 29.96 |
| CNF 0-shot + SMC (Lew et al. 2023), particle=8 | 26.27 | 04.07 | 03.14 | 19.85 | **100.0** | **100.0** | 27.51 |
| CNF 1-shot + SMC (Lew et al. 2023), particle=8 | 27.40 | 04.65 | 03.84 | 20.29 | **100.0** | **100.0** | 34.10 |
| CNF 2-shot + SMC (Lew et al. 2023), particle=8 | 28.44 | 05.93 | 04.54 | 20.70 | **100.0** | **100.0** | 48.76 |

Table 1: Performance results on generation quality, constraint satisfaction, and time over the CommonGen test set for different generation methods: *decoder only*, *prompt only*, and *prompt + decoder*. Results include a comparison against soft constraint decoding, i.e., beam-based NeuroLogic (NL), and hard constraint decoding, i.e., masked-based Sequential Monte Carlo (SMC). Two prompting strategies were conducted: abstract (ABS) and conjunctive normal form (CNF). Each prompting strategy leveraged 0-shot, 1-shot, and 2-shot in-context examples. With the exception of time (lower is better), a perfect score is 100.

inflections, one would expect higher constraint satisfaction across all experiments, however, this is not the case. With the exception of *CNF 1-shot*, *ABS* style prompting obtains higher satisfaction than *CNF*.

*ABS* prompting outperforms *CNF* prompting in terms of generation quality across all experiments. *CNF* style prompts are inherently more structured and further from 'natural language' compared to *ABS* style prompts. This suggests structured prompts are less beneficial and may require a fine-tuning strategy.

Increasing input length does not have significant impacts on inference time. Despite *ABS* prompts having a smaller constraint representation size than *CNF* prompts, there is little change in inference time across all *n-shot* experiments.

In-context examples boosts quality in both prompting strategies, but hurts satisfaction in *CNF 2-shot*. Including more than one in-context example worsens constraint satisfaction for *CNF* style prompts. This suggests that extending the input context with inflections for every in-context example may lead to noisy, sub-optimal distributions during generation.

**Decoder Only.** In this section we discover the impacts of the output layer, i.e., decoder, on generation quality, constraint satisfaction, and time. We compare two decoding strategies: soft constraint decoding, i.e., beam-based *NL* and hard constraint decoding, i.e., masked-based *SMC*.

In the absence of fine-tuning, beam-based/soft constraint decoding, i.e., NL, encounters challenges in both generation quality and constraint satisfaction. Compared with SMC, NL is more dependent on a high quality output distributions. This suggest that soft constraint decoding may require higher quality output distributions from the LLM.

Increasing the number of particles for SMC decoding does not yield quality or satisfaction improvements while increasing inference time. This observation indicates that the underlying distribution may be of low quality, as increasing the number of particles does not enhance performance. Moreover, in cases of uncertainty, the decoder will not see benefits by increasing computational resources.

Although the SMC decoder achieves 100% constraint sat-

isfaction, this achievement comes at the cost of significantly increased inference time. For example, the longest inference time recorded among the tested prompting strategies was only 1.85 seconds, in contrast, SMC with 8 particles required a considerably longer duration of 22.92 seconds for generation. This indicates a substantial increase in computational time required to achieve complete constraint satisfaction with masked decoding strategies, such as SMC.

Despite the improvements in constraint satisfaction, decoder only strategies tend to degrade generation quality and increase time.

**Prompt & Decoder.** Although prompt only strategies have higher performance on generation quality and time, they cannot provide any guarantees on constraint satisfaction. Conversely, decoding strategies optimize over constraint satisfaction, but at the cost of generation quality and time. In this section we aim answer whether the prompt and decoder can work together to improve the disadvantages of using the prompt or decoder alone, i.e., an end-to-end system. More specifically, we would like to understand how different strategies work together and whether they induce any trade-offs between our metrics.

Augmenting the prompt with constraints enhances generation quality and constraint satisfaction, indicating that prompting results in a higher-quality output distribution for the decoder to operates on. Notably, the *NL* decoder, although underperforming as a standalone decoder, shows remarkable improvement in quality when combined with prompts. This demonstrates that soft constraint decoder performance depends on the quality of the output distribution.

Although prompting improves quality in *SMC*, it has significant impacts on time. Checking for hard constraints within the *SMC* decoding strategy is less scalable when compared to the implementation of soft constraints in *NL*. In contrast, the *NL* decoder benefits in both quality and time. Due to the higher quality output distributions produced with prompting, the *NL* decoder spent less time searching, leading to reductions in inference time.

Across most experiments, the *NL* decoder achieves higher generation quality than *SMC*. The use of soft constraints in *NL* results in less drastic distribution changes compared to *SMC*, allowing for higher quality generation, albeit with a trade-off in constraint satisfaction.

Despite the *NL* decoder leveraging CNF formula, it exhibits higher satisfaction levels with *ABS* style prompts. This indicates that structured prompts could potentially limit the model's performance by producing sub-optimal output distributions for the decoder. It suggests that high-level concepts and relationships might be more effective inputs to the model when optimizing the output distribution for decoding.

## 7 Discussion & Future Work

Our results illustrate that we are not able to optimize over all metrics by using prompting or decoding alone. However, we can leverage the benefits of one strategy, e.g., prompting, to mitigate the risks of another, e.g., decoding, thereby finding a sweet spot between all metrics in an end-to-end system. In future work, we would like to extend current prompting and decoding methods to handle complex constraints, e.g., semantic constraints. Additionally, we would like to develop prompting techniques that result in optimal output distributions for the decoder to operate on.

**Challenges of Decoding with Semantic Constraints.** Extending current approaches (Lu et al. 2021; Lew et al. 2023) to handle semantic constraints poses challenges. Some domains may have many constraints, it might be too time-consuming to check to what degree a set of possible sequences satisfy all constraints in these domains during query time. One might check constraints in order of their importance or probability of being violated based on previous observations to save some time. We can also use current research on reasoning over constraints to find a minimal set of constraints that imply the entire set of reduce the number of constraints (Abiteboul, Hull, and Vianu 1994).

Another possible challenge is that one might have to check relatively long sequences of text to detect violations of semantic constraints. For example, the relationship between two entities might be represented in a relatively long sentence (paragraph) with each entity is placed in one end of the sentence (paragraph). The longer the size of examined sequences gets, the more generated sequences must be checked to compute the one with the largest posterior probability. This may significantly increase the time of returning an answer to the user. To speed up this process, one may test the constraints in the order of how close the mentions to the concepts or relationships usually appear in the text to prune some candidate sequences early. Another useful technique is to consider a relatively small sample of possible sequences instead of the entire set to return the final consistent result.

**Challenges of Prompting with Semantic Constraints.** Due to the uninterpretability of the LLM's learned representations, it is unclear whether prompting with constraints will result in consistent and accurate models. Recently, researchers have observed that explaining the properties of the desired answers gradually, i.e., Chain of Thoughts (CoT), improves the accuracy of answering questions over LLMs (Wei et al. 2023). One may use this property and provide the language model with step by step explanation of the modified prompt. To modify input prompt $P$, one should first identify the entities that appear in the prompt to find relevant constraints in $C$. Since constraints are usually written in subsets of first-order logic, one can express them in form of natural language, e.g., by translating $\subset$ to *is a subset of*. The final prompt will be a composition of the original one and natural language translation of its relevant constraints.

In cases where the modified prompt exceeds context length $k$, it may be necessary to employ constraint minimization and prioritization strategies. To minimize constraint representations, one may leverage the hierarchical structure of constraints , i.e., consolidate multiple constraints into broader constraints that imply them. To prioritize constraints, one technique is to weight constraints based on their coverage or relevance to the input prompt $P$. Another approach involves prioritizing constraints with more edges or connections, i.e. relationships, to other concepts.

# 8 Related Work

**Semantic Parsing.** LLMs have been an effective approach for program synthesis via semantic parsing (Poesia et al. 2022; Singh et al. 2023). These methods employ constrained semantic decoding techniques to guarantee that the generated output aligns with the syntax and grammar of the target programming language. This approach is similar to the use of lexical constraints in our work. However, unlike our work, semantic parsing does not address the complexities and linguistic coherence inherent in natural language. In our work, we aim to extend beyond simple lexical constraints to support more complex constraints.

**Self-Consistency of Language Models.** It is known that language models produce contradictory answers to the questions that seek the same information but phrased differently. Researchers have proposed methods to address this issue by prompting the language model to critique and refine its own output during inference (Madaan et al. 2023). This method prompts the language model with differently phrased questions and builds a (weighted) model over answers to infer the most likely result. We, however, mainly focus on ensuring that the language model follows semantic constraints.

**Extracting Knowledge from Language Models.** Researchers have proposed methods to extract generic statements or factual knowledge from language models using prompt engineering and human supervision (Bhagavatula et al. 2023). The prompts are constructed in a way that encourages succinct factual statements. They use human labeled data to detect inaccurate outputs and fine-tune the language model. It might be challenging to collect a sufficient amount of training data to extract accurate statements.

**Querying Language Models.** There has been some recent effort to design programming languages for prompting large language models, i.e., *language model programming* (Beurer-Kellner, Fischer, and Vechev 2023; Microsoft and Lundberg 2023; Computing and Louf 2023). There are generally domain-specific programming languages to extract information from and control the output of a large language model to satisfy the users' input hard constraints, akin to where conditions in SQL queries. Some of these languages resemble database query languages, e.g., SQL (Microsoft and Lundberg 2023). These languages aim at making it easier to query and prompt and optimize the number of calls to large language models. However, these languages do not generate consistent results conditioned on domain constraints. Thus, they may return answers that violate semantic constraints in the domain.

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

## A  Generated Sentence on Concept Set

In figure 1, we provide the sentence generated for a concept set in the CommonGen test set across all reported experiments in Section 6.

**Concept Set = [laundromat, book, laundry, read, wait]**

*Prompt Only*

[ABS 0-shot]: A boy read book in laundromat waiting for his laundry to finish.
[ABS 1-shot]: A boy in a book store waits to buy a book, now with my laundry, about to go to my laundromat.
[ABS 2-shot]: A boy waiting for a ride on the curb reads a magazine before putting the dusty suitcase inside a laundromat.
[CNF 0-shot]: A boy ia washed his his clothes that she has in the the laundromats for the laundry and to wait for the bus.
[CNF 1-shot]: A boy was reading books while sitting in a laundromatum waiting for laundry to be done.
[CNF 2-shot]: A boy who read books at a laundromats waited in a line to be the next to use the washers and dryers.

*Decoder Only*

[NL, beam=8]: A boy 12 years old, a girl 12 years old, a boy 11 years old, a girl 11 years old, a boy
[NL, beam=32]: A boy 16-year-old boy has been arrested on suspicion of murder after a 17-year-old boy was stabbed to death in
[NL, beam=64]: A boy 16-year-old boy has been arrested on suspicion of murder after a 17-year-old boy was stabbed to death in
[SMC, particle=8]: A boy buys books steals reading sheets his laundry emanates waiting in the laundromat his soda pointy h.
[SMC, particle=16]: A boy uses a book to light reading at a laundromat at a laundry structure identified waiting in the background as an Army Aviation Regiments
[SMC, particle=32]:A boy was in the waiting room and the books he was reading were the laundry drying laundromat and the man in the TV reef area

*Prompt & Decoder*

[ABS 0-shot + NL, beam=8]: A boy laundromat book laundry read waited.
[ABS 1-shot + NL, beam=8]: A boy laundromat book laundry read wait.
[ABS 2-shot + NL, beam=8]: A boy laundry his laundromat while waiting for his book to read.
[CNF 0-shot + NL, beam=8]: A boy \ud83d\udc66 \ud83d\udc66 \ud83d\udc66 \ud83d\udc66 \ud83d\udc66 \ufffd\ufffd
[CNF 1-shot + NL, beam=8]: A boy read a book in the laundromat while waiting for his laundry.
[CNF 2-shot + NL, beam=8]: A boy is reading a book in a laundromat.
[ABS 0-shot + SMC, particle=8]: A boy at the laundromat reads a book about laundries on his wait for his wait for the book to be laundered.
[ABS 1-shot + SMC, particle=8]: A boy in a laundry meets his laundromat friend who reads the book waiting for a wait that has been answered.
[ABS 2-shot + SMC, particle=8]: A boy waits at laundromat in front laundry with a book reading until it reads with the help of a waitress
[CNF 0-shot + SMC, particle=8]: A boy in a laundromat reads a book while he waits until a laundry to read another book.
[CNF 1-shot + SMC, particle=8]: A boy reads bill books at a laundromat as he waits to measure laundry in capsules.
[CNF 2-shot + SMC, particle=8]:A boy calls and waits at the laundromat to read the books on his laundry list before finally returning to wait.

Figure 1: Generated sentence sentence for concept set=[laundromat, book, laundry, read, wait] across different generation methods: *decoder only*, *prompt only*, and *prompt + decoder*. Two decoding strategies were used: soft constraint decoding, i.e., beam-based NeuroLogic (NL) (Lu et al. 2021), and hard constraint decoding, i.e., masked-based Sequential Monte Carlo (SMC) (Lew et al. 2023). Two prompting strategies were conducted: abstract (ABS) and conjunctive normal form (CNF). Each prompting strategy leveraged 0-shot, 1-shot, and 2-shot in-context examples. We begin the generated output sequence with "A boy".