# OpenReview forum: "Towards Consistent Language Models Using Controlled Prompting and Decoding"
_AAAI.org/2024/Workshop/NuCLeaR — NuCLeaR 2024_

### Official Review · Reviewer_L9VU · 2023-12-06
**The paper compares constraint satisfaction in LLMs and explores different decoding and prompting techniques that can help here.**

**Rating:** 7
**Confidence:** 4

**Review:**

Strengths:
1. Compares constraint satisfaction in LLM which is important for consistency.
2. Authors try both prompt and decoder-based techniques.

Weakness:
1. Literature review missing key papers
    a. Synchromesh - constrained generation for Code Generation (https://openreview.net/forum?id=KmtVD97J43e)
    b. Format5 - constrained generation and pretraining (https://arxiv.org/abs/2310.17306)
    c. DataVinci - constraint guided semantic data repair (https://arxiv.org/abs/2308.10922)
2. The paper does not introduce any ideas and is an amalgamation of existing techniques. The findings like in-context examples help improve performance are already well known.
3. The paper should have more examples of success and failure cases for different techniques. Especially since it is an analysis paper.

Questions:
1. What was the prompt used for decoding strategies. Based on the poor result on standalone and improvement with prompting it looks like the prompt is to blame.
2. It would be interesting to see if finetuning can help here.

Formatting
1. related work section has a todo in header
2. missing related work as highlighted in weakness
3. for the dog example from commonGen, it would be good to show an example of a high score result.

---

### Official Review · Reviewer_pyWQ · 2023-12-07
**Meaningful paper and interesting research topic**

**Rating:** 6
**Confidence:** 4

**Review:**

Pros:
1. This paper address a meaningful and interesting research question that the existing large language models often return incorrect and inconsistent answers to input questions.
2. This paper has conducted solid experiment and the detailed experiment result.

Cons:
1. It will be better to give an concrete example for Declarative Constraints.
2. Although the experiment result is solid, only LLaMA-2-7B is used which will be good if there is more backbone models been considered.

Typo issues:
1. Related Works TODO: Reword -> Related Work

---

### Official Review · Reviewer_WZes · 2023-12-08

**Rating:** 5
**Confidence:** 3

**Review:**

**Summary**

The paper explores approaches to minimize the inconsistencies in the LLM generated texts without fine-tuning. Authors discuss some approaches for representing semantic and lexical constraints and how to enforce these constraints during text generation. Experiments on CommonGen dataset shows the benefit of combining prompt engineering and constrained decoding techniques for generating text in LLMs that meet provided requirements

**Strengths**

1. Interesting discussion on enforcing semantic and lexical contraints at various stages of text generation without requiring the computationally expensive fine-tuning
2. Insightful ablation studies on CommonGen dataset

**Weaknesses**

1. The contributions of the paper are not expressed well in Sections 2, 3 and 4
2. The experiments conducted in Section 5 do not cover the ideas discussed in previous sections especially semantic constraints or repairing output sequences
3. Proposed high-level declaratives constraints (prompt engineering) and combining that with decoding techniques is not technically novel

---

### Decision · Program_Chairs · 2023-12-11

Accept